# The Influence of Diverse Cultures on Nutrition, Diabetes Management and Patient Education

**DOI:** 10.3390/nu16213771

**Published:** 2024-11-02

**Authors:** Jessica Shapiro, Martin M. Grajower

**Affiliations:** 1Montefiore Medical Center, Bronx, NY 10467, USA; jessicaleighshaffer@gmail.com; 2Division of Endocrinology, Albert Einstein College of Medicine, Bronx, NY 10463, USA

**Keywords:** nutrition, diabetes, cultural foods, plate method, motivational interviewing, religious observances, diabetes education

## Abstract

**Background/Objectives**: Providing relevant, patient-centered care starts with recognizing that patients living with diabetes are racially and ethnically diverse, which will influence their dietary behaviors. **Methods:** The authors draw upon literature descriptions and personal experience in clinical practice dealing with ethnically diverse populations and include guidance offered in the literature regarding how to address these unique aspects when managing and educating patients with diabetes. **Results**: Proper interviewing techniques are described when dealing with culturally diverse populations, including ascertaining cultural, religious, and ethnic influences on dietary choices, and advice is given on how to improve nutritional behavior in these patients while acknowledging and validating these influences. **Conclusions**: When a proper nutrition interview is conducted, such as using motivational interviewing, aspects of the patient’s cultural, religious, ethnic, and other influences can be ascertained, and appropriate advice can be given to the patient on how to modify these influences to achieve a healthier nutritional behavior.

## 1. Introduction

Providing relevant, patient-centered care starts with recognizing that patients living with diabetes are diverse and, therefore, one cannot have a “one-size-fits-all” approach to diabetes counseling and education. Understanding the biological elements that influence the course of diabetes is important, but it is just one factor when aiming to improve diabetes-related outcomes with culturally relevant nutrition education. Race and ethnicity are socially constructed terms that are used for classifying some study populations; these classifications cannot be relied upon as the sole biological explanation to disease incidence, prevalence, and outcomes.

Understanding the biological elements that influence the course of diabetes is important but it is only one factor when aiming to improve diabetes-related outcomes with culturally relevant nutrition education.

A person’s dietary pattern and lifestyle are often associated with cultural patterns and contribute to self and communal identities. They are heavily influenced by a complex mix of a one’s own lived experiences along with social determinants that include cultural background, family and social networks, religious practices, education, health literacy, perception of body image, food access and availability, and the broader environment.

In this paper, we review some of the literature and use personal clinical experience to describe how different cultural traditions can influence diabetes management and education.

## 2. Methods

The South Bronx is known to be one of the poorest neighborhoods in New York City. Its population is made up of a majority of high-risk minorities, including African Americans, Latinos, and Asian Americans [1,2,3,4]. Neighborhoods with the highest poverty also have some of the highest diabetes rates across the city; diabetes is nearly 70% more common than in low-poverty neighborhoods. Living in economically disadvantaged communities makes it more difficult to access healthy foods and exercise, contributing to the (not well-understood) disparity in diabetes prevalence. This article uses as an example the experience of a registered dietitian and certified diabetes educator (JS) practicing in the Bronx for the past 15+ years. The Bronx, one of five boroughs in New York City, has the highest prevalence of diabetes in New York state.

## 3. Results

### 3.1. The Connection Between Culture and Cuisine

While every native culture includes a set of behaviors and food practices learned at home, one must also consider how these traditions have been influenced and adapted by the amalgamation of the mainstream American diet. This acculturation, or adoption of, beliefs and customs of a new culture, such as with mainstream culture, is associated with a higher fat intake, lower fruit and vegetable consumption, and an introduction of more processed foods and sweets [5].

A study of Latinos was able to demonstrate that the “greater years living in the United States was associated with lower dietary pattern healthfulness”, which hints to the influence of acclimation to the local dietary pattern. Having just arrived in the United States versus being a second-generation immigrant family may also have different implications, based on the extent of “dietary acculturation”. According to Maldanado et al., “dietary acculturation [is] a complex and dynamic process by which immigrants typically adopt the cultural practices of the host country and abandon the cultural dietary choices and behaviors practiced in the country of origin”. That shift from a traditional dietary pattern to one reliant on energy-dense and less nutrient-dense foods (such as burgers, fries, and soft drinks) generally translates to an increase in chronic-disease risk [6].

#### Barriers to Seeking Nutritional Care

There are a number of barriers to providing competent nutrition education when dealing with a diverse population. Some of these, and possible remedies, are summarized in Table 1.

Patients from diverse cultural backgrounds often carry with them their own set of beliefs, spoken languages, decision making practices, and communication styles. Even with all the unique characteristics of our patients, there are universal challenges to consider in order to break down the barriers to ensure understanding and adherence, leading to high-quality care and improved outcomes. Strategies for an inclusive environment include having a diverse workforce and adequate translation services available for patients. There is a significant difference when working with patients using the language they are most comfortable with or having an in-person translator or video conference versus voice-only sessions, due to the importance of non-verbal cues.

It is also not always just about the patient. Many cultures are family-focused, and patients may live with extended family. The family dynamics must be considered when managing diabetes. One needs to verify who is the primary party responsible for the household food shopping and cooking. If it is not the patient themself, then with their consent, it is valuable to have the patient accompanied by any other family members who are actively involved in their eating patterns [8].

It is imperative to consider that feelings about weight and body image can vary between cultures. Some cultures place a higher value on, and favor, women with larger bodies. Therefore, the discussion of weight loss goals “should consider a patient’s individual perception of body image”. In such situations, one should emphasize health gains and not weight loss. For example, instead of focusing on the weight loss aspect, one should explain that eating healthfully and moving more will lower their blood glucose levels [4].

Traditional health beliefs and fears, and dietary customs, may be deeply embedded in the patient’s culture. The media may also have a large influence on behaviors. The current marketing strategies towards consumers have moved to another level due to social media. Two common phrases to listen for are “TikTok told me” or “I saw it on Instagram”. Providers should keep a pulse on the food trends being shared across social media platforms, which are both helpful and harmful for people living with diabetes, and be ready to dispel misconceptions and myths.

### 3.2. Successful Outcomes Start with Building Rapport

#### 3.2.1. The Value of Internal Motivation

Establishing an open dialogue and partnership with patients can begin from the start of the relationship by meeting them “where they currently are at,” in respect to both their culture practices and their personal health journey. Does the patient know why they were referred for nutrition counseling in the first place? Do they know how their current diet and lifestyle may be affecting their health? One can gain perspective by encouraging each patient to share their abbreviated personal health journey and what motivates them to make changes. Patients who are able to locate internal motivation find the most success in making healthy lifestyle changes.

#### 3.2.2. Developing Cultural Humility

While it is helpful to have a baseline awareness of and sensitivity to the population in a practice or clinic, healthcare providers need to do more. The Center for Disease Control explains that it is essential to embody cultural humility, which involves an “ongoing process of self-reflection” and a “non-judgmental willingness to learn” in order to “gain a deeper realization, understanding, and respect of cultural differences”. Having cultural humility is an inquiry mindset that recognizes that the patient is the expert [9].

One can be inquisitive without judgment and be mindful not to demean cultural staples. Shaming creates stigma and could inherently affect a patient’s self-worth. So much can be learned about foods and recipes from all over the world by asking questions. Many questions will help enlighten the provider to clarify what is in a patient’s report. Examples are the following: “How do you cook that? What ingredients are in that recipe? Is that a meal that is home-cooked by you or a family member? What type of restaurant is that from? Do you add anything to your coffee/tea? What are your favorite foods? How many times per week do you eat your favorite foods?” There is nothing wrong with looking up a dish or food item together with a patient to ensure the provider and patient are referencing the same thing.

### 3.3. Assessing Dietary Intake

Evaluating a patient’s baseline dietary habits and assessing the extent of dietary acculturation can be done by guiding patients through a 24 h dietary recall, or by asking for their usual daily food intake. This tactic aims to obtain a general idea of what a patient consumes on a typical day, while also providing ample opportunities to become familiar with a patient’s culture and their cuisine.

#### Uncover the Reason(s) Why Certain Foods Are Not Eaten

It is equally valuable to understand why certain foods, such as fruit and vegetables, are not eaten. Verify if these reasons are justified concerns or perceived constraints. Often, they are likely multifaceted.

In terms of socioeconomics, there may be a financial cost of purchasing the food, as well as a time burden necessary for food preparation. These challenges can be overcome by having resources on hand for food assistance, such as teaching a patient how to buy non-perishable goods in bulk or providing a referral to a local food bank, or time-saving ideas, such as teaching the art of meal planning and/or integrating ready-to-eat foods.

Thoughts commonly shared are as follows: “The fruit and vegetables taste horrible in America. Back in my country they were so much better that I don’t want to eat them here” or “I didn’t eat fruit and vegetables as a child. My mom never cooked them”. One could allow space for the patient to express their food sentiments and then validate their past experiences. This allows for assessing the patient’s readiness to make a change.

### 3.4. Nutrition Education Considerations

After fulfilling the groundwork of understanding a patient’s current lifestyle habits and behaviors, the next part is confirming what a patient is *actually* willing to do to make healthful, yet sustainable, changes. One should focus on having the patient identify what subtle changes they feel confident in implementing. Then, these efforts can be supported with resources and guidance, as requested and directed by the patient. Studies confirm that starting with making one small change has the ability to create a trickle-down effect, leading to a larger transformation [5,6,8,9].

#### 3.4.1. Clear Communication

It is important to make note of a patient’s education level and literacy. This may require patience with accents and possibly the need for an interpreter when there are language barriers. If a person is unable to read, they may be embarrassed to admit it. One could test the patient’s competence by having them teach back, or summarize, what they learned.

#### 3.4.2. Cook More

In theory, one of the most powerful ways to immediately improve a patient’s diet is to eat out less and cook more at home [10]. In reality, this may be simpler said than done. Not everyone can cook. It is commonplace to hear from patients that they grew up being told to “get out of my kitchen” instead of being invited to cook together. A good opening could be “What is your comfort level around cooking?” One should aim for the patient to take one step away from eating out all the time in favor of preparing meals at home. For example, if a patient currently eats every meal out of the home, one could try to push them to have one meal a day at home. The availability of ingredients, skill level, kitchen access, and interest level may all impact the probability of a patient being able to actually cook more at home.

#### 3.4.3. Incorporating Familiar Foods into a Healthy Plate Template

The “Plate Method” is a fool-proof way to educate patients on a balanced diet with proper portions, while allowing for flexibility for the types of foods to be incorporated. Many of the groups residing in the Bronx have a carbohydrate-dominant dietary pattern. Photos of different cultural food plates following these parameters are helpful.

This can be implemented by dividing the plate into three parts. Mixed meals or combination meals, such as with casseroles, soups, etc., can still follow this guide when preparing and proportioning the food:(1)Fill up half the plate with non-starchy vegetables (e.g., salad, green leafy vegetables, okra, broccoli).(2)Fill-up one-quarter of the plate with lean protein (e.g., meat, fish, eggs, plant-based proteins like tofu).(3)Fill-up the last quarter of the plate with carbohydrates (e.g., grains, starchy vegetables like potatoes, fruit, yogurt, beans).(4)Complete the meal with a zero- or low-calorie beverage, with water recommended as the beverage of choice.(5)Highlight the importance of limiting saturated fat sources (from animal products) and suggest small amounts coming from unsaturated sources such as olive oil, avocado oil, nuts, and seeds [11,12].

#### 3.4.4. Modifications to Meals and Recipes

It is possible to teach patients how to make small changes in traditional foods and flavors in order to increase the health value of a meal or recipe. A curry from Jamaica versus a curry from India may have different ingredients. Even within the same country there may be significant differences in how the supposedly same dish is prepared. One should ask about the ingredients and how the recipe is cooked rather than make assumptions. The following are some suggestions that patients can apply in their own home:

Add non-starchy vegetables in any way possible. Encourage patients to try adding a chopped salad to a meal, or incorporate more non-starchy vegetables to soups, stews, or curries. A bag of frozen spinach is good to keep on hand.

Consider the type and portions of carbohydrates. Carbohydrate-centric cultures may include refined grains and multiple types of carbohydrates in each meal, such as rice and beans, rice and pasta, or rice and roti, pasta and bread. Patients can be asked to consider cutting down the portion size (sticking to one-quarter of the plate); picking just one carbohydrate source (selecting rice OR roti, not both); replacing some or all of the refined grains with whole grains (trying brown rice instead of white rice); looking at other ingredients (e.g., cutting back on added butter/oil used in cooking and/or as a condiment); and/or incorporating vegetables to add volume (e.g., cutting the rice portion in half and adding riced cauliflower).

Look at the types of fats and healthier swaps. For example, the fats used when cooking a recipe (e.g., exchange lard with olive oil), or the condiment/topping added to a dish (e.g., avocado instead of bacon as a breakfast side).

Be on the lookout for sources of added sugar and discuss ways to decrease it. Sources may include sugar-sweetened beverages like soda, store-bought and homemade juices/smoothies, or sweetened/condensed milk.

#### 3.4.5. Familiarize with the Local Food Environment

In parts of the Bronx, bodegas outnumber supermarkets twenty to one, with limited healthy options. Neighborhoods in the Bronx commonly include the traditional food carts but will also have carts with cultural foods readily available. For example, in the summertime, one encounters push carts selling a frozen sorbet-like treat, called *helados*. While cheap, refreshing, and delicious, *helados* sold on the street do not have a nutrition facts label and are high in sugar [13].

The nutritionist can scope out the local neighborhood for healthier options. Some of the best sources for finding hidden gems could be from other patients or community resources. Examples would be locating fresh fruit and vegetable stands or a farmer’s market, or a local fish market that will steam fresh fish for a workday lunchtime special. Montefiore Health System in the Bronx has been coordinating a variety of efforts to improve access to healthy foods. Initiatives available to patients include having a healthy and affordable meal in the hospital cafeteria or going to one of the local bodega partners that has healthier offerings [14]. If a patient has a favorite restaurant, one can review the menu with them to help them identify the best choices.

#### 3.4.6. It Is Not Just About Food

Managing diabetes requires a whole lifestyle approach, and talking to patients about adding physical activity should come together with discussing food choices. Using the term “exercise” may be a deterrent, as it comes with a stigma of having an expensive gym membership or a rigid, time-consuming routine. On the other hand, the term “physical activity” allows for a more generous interpretation of movement, making movement more accessible and available. This discussion can start with asking patients what they like to do for movement and inquiring about past participation in any types of sports or activities that they enjoyed. According to the World Health Organization’s “Guidelines on Physical Activity and Sedentary Behaviour”, the best form of activity is absolutely anything the patient will do, and any amount is better than none. Start with small amounts of physical activity and then gradually build up from there. Studies have shown that having the patient wear a pedometer, or using an application on a smartphone, helps motivate the patient to be more physically active [15].

### 3.5. Religious and Spiritual Influences on Diet

Family celebrations and religious holidays are full of food and considerable temptation, so patients will benefit from learning how to approach these scenarios. There may also be some implied expectations that can impact a person’s food choices at these gatherings. There tends to be a consensus around fearing they will disrespect an elder or another family member, which may be implicated from not finishing their meal, or not accepting food. A ubiquitous expectation heard from many patients is the importance of finishing their plate, as they were raised to feel guilty if their plate was not finished. This ideal is shifting, but it is still present and ingrained in society. This becomes even more of a challenge in patients on GLP-1 RA (glucagon-like peptide-1 receptor analog) medications, where these medications cause delayed gastric emptying, resulting in early satiety. “Finishing one’s plate” with those last few bites may be the difference between feeling satiated or becoming nauseous.

Everyday customs around food may or may not be swayed by a patient’s spirituality or religion. There are some prominent religious and spiritual practices that affect the types and quantity of food that are consumed.

#### 3.5.1. Religious Fasting

Religious fasting is a spiritual practice observed by various faiths. The act of abstaining from all foods and beverages, or just certain ones, takes place over a set period of time. During these times, it is essential that patients living with diabetes work with their provider to establish an action plan regarding making changes in oral medication or insulin. In addition, they must carefully monitor their symptoms and blood sugars via a continuous glucose monitor or by fingerstick throughout, since there is a risk of dehydration and/or hypoglycemia [4,5].

Patients may not bring up this subject on their own and may rely on their own instinct or the internet to manage their diabetes while fasting. It therefore is a good idea for the provider to be aware of upcoming religious fast days and to proactively discuss management with the patient.

#### 3.5.2. Ritual Foods

Some holiday rituals include special foods/beverages that are only consumed at certain times of the year. It is a good idea to have an open conversation about the significance of these customary foods and the frequency that the patient consumes them throughout the year. This is pertinent information for guiding patients on making modifications that will both support the patient from a diabetes management perspective while conforming to their expressed preferences.

For example, in Judaism, every Sabbath and on holidays, an egg- or water-based bread called challah is eaten. If a patient consumes multiple slices of this bread on a regular basis, it may be impacting their blood sugars. Strategies might include cutting back on the challah by eating smaller amounts, electing to omit other carbohydrates during the mealtime, and/or switching to whole grain challah.

### 3.6. How You Ask Questions Is Just as Important as What Questions Are Being Asked

Patients have to be internally motivated and willing to make changes. A proven approach is to use motivational interviewing (MI) to empower the patient and to guide each session, rather than preaching to the patient and *telling* the patient how to change their lifestyle. MI is an empathetic, “person-centered counseling style for addressing the common problem of ambivalence about change”. The core concepts of MI are identified in the acronym “OARS”: [16,17].

#### 3.6.1. Asking Open-Ended Questions

What is it that brings you here today?

Tell me more about…

Can you walk me through a typical day of eating?

Which foods do you traditionally eat during holidays and special occasions?

#### 3.6.2. Affirming (Genuine Appreciation- Emphasize Client Strengths, Successes, and Efforts)

You took a big step in coming here.

You’ve been checking your blood sugars daily and taking your meds consistently; it shows a real investment in your health and future.

Despite the challenges that you just told me have come up with making healthy choices, you have come back today, which shows that you really understand the importance of making changes to your diet.

#### 3.6.3. Reflective Listening

What I hear you saying is you understand that losing weight may play a role in controlling your blood sugars, but you are very comfortable in your skin and don’t want to look malnourished.

I get the feeling there is a lot of pressure on you to change, and you are not sure you can do it because of difficulties you had when you tried in the past.

#### 3.6.4. Summarizing

You enjoy eating vegetables and really want to start including them in your diet on a regular basis, but you have identified some barriers: you don’t do the grocery shopping and fresh vegetables are not a regular item purchased; your wife does the cooking and isn’t used to cooking vegetables; and you don’t know how to address this issue with your wife. Did I miss anything?

## 4. Outcomes

A variety of randomized controlled trials analyzed if culturally tailored implementation strategies ultimately made a difference to patient outcomes. The efforts appear to be effective in improving glucose management and metabolic markers, as demonstrated in a review of these interventions for patients living with diabetes. Remarkably, improved barriers to care were also addressed with these interventions, including “educational knowledge gaps, language, low health literacy, and the role of family” [18].

## 5. Limitations

Our practice in a multicultural population in a dense urban city with a high incidence of diabetes offers us a unique perspective of cultural influences in diabetes management and patient education. Nevertheless, the limitations of this paper include being exposed to a limited number of religious, ethnic, and racial populations; thus, the findings may not be applicable in certain medical practices. Also, we do not have access to hard data to quantitate how much of our experience is influenced by socioeconomic forces.

Furthermore, we do not have data to prove that the techniques mentioned resulted in better patient outcomes. Lastly, we have not attempted in this paper to fully assess and describe social media and technology influences on nutrition.

## 6. Conclusions

Nutrition education for people living with diabetes should consider the patient’s cultural values, as well as social, environmental, and personal preferences. Working with a diverse population with unique characteristics in patients with diabetes is a challenge. Empowering the patient by eliciting their internal motivation, and, ultimately, recognizing and appreciating a patient’s individual connection to their cultural foods, yields better compliance and improved outcomes. Future controlled studies in culturally diverse populations employing the techniques outlined would be welcome, to improve diabetes care worldwide, relying more on nutrition and less on medication.

## Figures and Tables

**Table 1 nutrients-16-03771-t001:** Common barriers to providing competent nutrition education to a diverse population and approaches for patients living with diabetes.

Barrier	Approach to Barrier
Patient’s native language is not English	Having a diverse workforce. Ongoing training and evaluation on cultural competency. Collect data on patients’ primary languages and ensure adequate translation services are available, as needed [7].
Attitudes about weight and body image may not align	Focus on health gains, not weight loss [4]
Differences in communication styles	Utilize motivational interviewing to empower the patient and to guide each session, rather than preaching to the patient and telling the patient how to change their lifestyle.
Low literacy levels	Keep education material simple. Evaluate where a patient is at by testing their competence, having them teach back, or summarize, what they learned.
Patients not knowing how to access nutrition counseling/classes	Use different advertising methods, including mailing, printed materials, and email. Send calendar invites.
Worried about judgement	Employ non-judgmental techniques during interviews.
Reluctance to take diabetes medications	Emphasize nutrition first.

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
