# Peer review of "The Influence of Diverse Cultures on Nutrition, Diabetes Management and Patient Education"

_nutrients, 2024, doi:10.3390/nu16213771_

Round 1
Reviewer 1 Report
Comments and Suggestions for Authors
The authors presented an interesting review about the relationship between dietary behaviours and cultural and social aspects. The paper is interesting, but a more structured narrative review should be implemented.
The title "cultural foods" does not match the content of the article. Race and ethnicity are structural variables that should be considered
It seems an expert opinion rather than a review.
Other authors treated this topic:
- Christopher, Duggan., Christopher, Duggan., Anura, V, Kurpad., Fatima, Cody, Stanford., Bruno, F., Sunguya., Jonathan, C., K., Wells. (2020). 2. Race, ethnicity, and racism in the nutrition literature: an update for 2020. The American Journal of Clinical Nutrition, doi: 10.1093/AJCN/NQAA341
Tiffany, L., Carson., Michelle, I., Cardel., Takara, L., Stanley., Steven, K., Grinspoon., James, O., Hill., Jamy, D., Ard., Elizabeth, J., Mayer-Davis., Fatima, Cody, Stanford. (2021). 4. Racial and ethnic representation among a sample of nutrition- and obesity-focused professional organizations in the United States. The American Journal of Clinical Nutrition, doi: 10.1093/AJCN/NQAB284
Jaapna, Dhillon., Ashley, Jacobs., Sigry, Ortiz., L. Karina, Diaz, Rios. (2022). 8. A Systematic Review of Literature On the Representation of Racial and Ethnic Minority Groups in Clinical Nutrition Interventions. Advances in Nutrition, doi: 10.1093/advances/nmac002
B. Bernstein. (2022). 9. Nutrition interventions addressing structural racism: a scoping review. Nutrition Research Reviews, doi: 10.1017/s0954422422000014
Moreover, supplemental nutrition is another great topic that should be included [Simon, F., Haeder., Donald, Moynihan, 2023]. 12. Race And Racial Perceptions Shape Burden Tolerance For Medicaid And The Supplemental Nutrition Assistance Program. Health affairs Web exclusive, doi: 10.1377/hlthaff.2023.00472] because of its impact on several pathologies [10.1016/j.jnha.2024.100256].
I suggested authors choose appropriate MesH terms and perform a structured literature search.
Author Response
The title "cultural foods" does not match the content of the article. Race and ethnicity are structural variables that should be considered
We have revised the title as suggested.
I suggested authors choose appropriate MesH terms and perform a structured literature search.
We reviewed the literature as suggested and added references. Appreciate the suggestions and modified the paper accordingly.
Reviewer 2 Report
Comments and Suggestions for Authors
This is my review on your article “Cultural foods and Influences on Nutrition and Diabetes”.
Overall it is well-written. I would suggest these improvements:
· The abstract needs to be more specific in terms of methodology findings.
· You should avoid certain generalizations like the discussion of cultural practices without supporting data. In several claims, authors should add citations.
· You should create a clear distinction between different sections. Consider adding subheadings.
· Authors must acknowledge the limitation that this is a study of certain populations and globally applicable.
· Certain terms need better clarification (cultural humility, dietary acculturation etc.).
· The technology sector should be further analyzed.
Comments on the Quality of English LanguageEngilsh language is overall fine. For minor improvements, you should try to simplify certain sentences and terms.
Author Response
The abstract needs to be more specific in terms of methodology findings.
Abstract was modified with methods section added.
You should avoid certain generalizations like the discussion of cultural practices without supporting data. In several claims, authors should add citations.
We clarified when our comments were based on personal experience and added supporting references when appropriate and available.
You should create a clear distinction between different sections. Consider adding subheadings.
We added subsections and headings as suggested.
Authors must acknowledge the limitation that this is a study of certain populations and globally applicable.
We added a paragraph highlighting the limitations of our paper.
Certain terms need better clarification (cultural humility, dietary acculturation etc.).
In the relevant paragraphs we clarified these terms.
The technology sector should be further analyzed.
Our experience with the technology sector is limited. This suggestion is a good one but beyond our experience. We did add a few comments in the paper, and also indicated that more on technology would have been a worthwhile addition to the paper but beyond our experience.
Reviewer 3 Report
Comments and Suggestions for Authors
The authors provide an article on the issue of ethnically diverse populations and guidance offered in the literature on how to address these unique aspects when managing patients with diabetes. It was found that when a proper nutrition interview is conducted, such as using motivational interviewing, aspects of the patient’s cultural, religious, ethnic, and other influences can be ascertained, and appropriate advice can be given to a patient on how to modify these influences to achieve a healthier nutritional behaviour. This is interesting work. Appropriate methodology has been employed and the conclusions appear to be justified based on the data at hand. I have some recommendations for consideration.
1. Introduction. Is there a clear hypothesis that the authors are testing in their study?
2. Discussion. The authors need to consider the role of cultural and language barriers in seeking advice on nutritional approaches for the management of diabetes.
3. Discussion. The authors need to emphasize and elaborate on the novelty aspect of their work.
4. Discussion. The higher incidence of diabetes in these regional ethnic populations may just be a reflection of the poor socioeconomic status?
5. Discussion. The authors need to provide some data on the effectiveness of the described approaches.
Author Response
Introduction. Is there a clear hypothesis that the authors are testing in their study?
Introduction was modified to be clearer what our objective is in publishing this paper.
Discussion. The authors need to consider the role of cultural and language barriers in seeking advice on nutritional approaches for the management of diabetes.
We added comments regarding language barriers.
Discussion. The authors need to emphasize and elaborate on the novelty aspect of their work.
We emphasized the novelty aspect of our work in the discussion.
Discussion. The higher incidence of diabetes in these regional ethnic populations may just be a reflection of the poor socioeconomic status?
We added a comment on the socioeconomic status in the "limitations" section.
Discussion. The authors need to provide some data on the effectiveness of the described approaches.
Unfortunately we don't have hard data to provide. We noted this in the "limitations" section of the paper. We hope the reviewer still finds the paper worthwhile as it offers many suggestions for how to deal with diverse populations. Hopefully, others will use these suggestions to do a controlled study on their effectiveness.
Round 2
Reviewer 1 Report
Comments and Suggestions for Authors
The paper is improved.